# The Scarcity of Literature on the Psychological, Social, and Emotional Effects of Gastroparesis in Children

**DOI:** 10.3390/children7090115

**Published:** 2020-08-31

**Authors:** Tiffany H. Taft, Bethany Doerfler, Emily Edlynn, Linda Nguyen

**Affiliations:** 1Division of Gastroenterology & Hepatology, Northwestern University Feinberg School of Medicine, Chicago, IL 60611, USA; bethany-doerfler@northwestern.edu; 2Pediatric Behavioral Medicine, Oak Park Behavioral Medicine LLC, Oak Park, IL 60301, USA; eedlynn@opbmed.com; 3Division of Gastroenterology, Stanford University, Palo Alto, CA 94305, USA; nguyenlb@stanford.edu

**Keywords:** gastroparesis, anxiety, depression, health-related quality of life, PTSD

## Abstract

Gastroparesis (GP) is a chronic, gastric dysmotility disorder with significant morbidity and mortality. The hallmark of GP is the delayed emptying of the contents of the stomach in the absence of any mechanical obstruction. Patients most commonly report chronic symptoms of nausea, vomiting, feeling full quickly when eating, bloating, and abdominal pain. Treatments are limited with relatively poor efficacy. As such, children with GP are at significant risk for the development of psychological co-morbidities. In this paper, we provide a topical review of the scientific literature on the psychological, social, and emotional impacts of gastroparesis in pediatric patients. We aim to document the current state of research, identify gaps in our knowledge with appropriate recommendations for future research directions, and highlight the unique challenges pediatric patients with GP and their families may face as they manage this disease. Based on the current review, research into the psychosocial impacts in children with GP is essentially non-existent. However, when considering research in children with other chronic digestive diseases, children with GP are likely to face multiple psychosocial challenges, including increased risk for anxiety and depression, stigma, and reduced quality of life. These significant gaps in the current understanding of effects of GP across domains of childhood functioning allow for ample opportunities for future studies to address psychosocial outcomes.

## 1. Introduction

Patients living with poorly understood chronic digestive conditions make up a large percentage of those seeking gastroenterology services across the world in both pediatric and adult practices. These conditions also come with considerable psychological morbidity and negative effects on quality of life. In this space, most attention has been paid to disorders of gut–brain interaction (DGBI), formerly known as functional gastrointestinal disorders (FGIDs). Gastroparesis (GP) is a chronic gastric dysmotility disorder affecting the emptying of the stomach [1]. Who is affected by GP appears to shift over the lifespan, with rates alarmingly higher among adult females. Current estimates suggest that GP impacts approximately 38 women and 10 men per 100,000 people in the United States [2]. In pediatric populations, no prevalence data exist as of 2020. However, males tend to represent younger patients with GP, while females predominate adolescent cases [3]. 

Gastroparesis has significant morbidity and mortality in adult populations, the latter of which distinguishes it from DGBIs, typically considered to be benign. Specifically, in one of the only epidemiological studies of GP, again in adults, one third of patients died, while an additional one third required hospitalization, medication, or tube feeding [4]. As GP is a chronic condition, children and adolescents diagnosed with GP are at risk for these outcomes as they enter adulthood. The associated costs of pediatric GP are substantial and increasing, with a 5.8-fold increase found between 2004 and 2013 (USD 6 million to USD 35 million) [5]. Rates of hospitalizations also increased significantly during this time, from 252 in 2004 to 1310 in 2013; unique patients increased from 174 to 723, suggesting an increased incidence of this disease among children. Estimates as of 2020 suggest that these numbers grossly underrepresent the true prevalence of pediatric GP, partially due to a lack of standardized diagnostic criteria [6].

In GP, the stomach fails to empty in an expected amount of time (typically under 4 h), which results in chronic nausea, vomiting, early satiety, failure to thrive, abdominal fullness and bloating, and pain [1]. The gold standard test for GP is the 4-h gastric emptying scintigraphy; however, significant limitations to this test exist, underscoring the need for updated diagnostics in both pediatric and adult GP patient populations [7]. Symptoms vary by age, with younger children experiencing more vomiting whereas adolescents report greater nausea and abdominal pain [3]. Endoscopic examination of the esophagus, stomach, and first part of the small intestine yields no abnormalities or mechanical obstructions to explain this delayed gastric emptying. Gastroparesis can have multiple causes, but the most typically found are post-viral infection and idiopathic, or of unknown cause [7]. Other, less common, reasons GP may develop in children include medications and certain types of surgery. Treatments are limited to certain pro-motility or anti-nausea medications [1,3], experimental surgeries [8], or gastric electrical stimulation implants [7,9]. Younger children tend to respond better to treatment, with up to 60% achieving symptom resolution after 3 years [3]. However, as patients age or have longer symptom durations, the efficacy of the available treatments is limited, leaving many patients continuously symptomatic [1,10,11]. This combination of symptom chronicity and low treatment efficacy has clear implications for negative psychosocial outcomes in this patient population. In addition to pharmacological interventions, dietary therapy is often offered as a treatment option in GP. In many cases, parents may prefer dietary changes over medication as these are often seen as less of a risk [12]. Diet modifications including low fat and fiber foods, small frequent meals, and liquid supplementation are often prescribed but with poor improvement in symptoms [3].

While the severity of GP is being recognized in the medical community, significant gaps remain in the literature regarding our understanding of the psychosocial effects of this disease when compared to similar literature across the digestive disease spectrum. Three reviews of gastroparesis in children are published at the time of this review, with limited to non-existent discussion of the psychosocial effects of GP [6,7,13]. Remarkably, the majority of research in pediatric GP is retrospective and only 12 randomized controlled trials were identified in the 2020 review by Kovacic and colleagues [7]. Based on this limited state of the research to date, we aim to generate a topical review of the literature on the potential psychosocial effects of GP to include: anxiety, depression, post-traumatic stress, social relationships, including stigma and withdrawal, and health related quality of life. We also aim to provide suggestions for future lines of research. 

## 2. Methods

A topical review of studies published up to May 2020 (no start date) in which pediatric participants with a confirmed diagnosis of gastroparesis was performed via the following online databases: Medline (PubMed), PsycINFO, and Google Scholar. Figure 1 represents the PRISMA guidelines used to select articles to be included. 

### 2.1. Inclusion Criteria

We included all studies that met the following criteria: (1) published in English in a peer-reviewed journal; (2) participants had either idiopathic, diabetic, or post-viral GP; (3) participants were ages 0 to 17 years; and (4) anxiety, depression, stigma, post-traumatic stress, and HRQOL were evaluated using standardized assessment tools. 

### 2.2. Procedures

Articles were identified and stored in an EndNote library. Titles or abstracts were returned via the following keyword search combinations applied separately to each of the identified databases above: children, adolescents, pediatric, gastroparesis, idiopathic gastroparesis, diabetic gastroparesis, post-viral gastroparesis, anxiety, depression, psychosocial, psychological, mental health, stigma, health related quality of life. Keywords were combined using “OR” statements with MeSH and tiab field codes, then grouped under the following concepts:Concept 1“Gastroparesis” OR “idiopathic gastroparesis” OR “post-viral gastroparesis” OR “diabetic gastroparesis.”Concept 2“Psychosocial” OR “psychological” “anxiety” OR “depression” OR “mental health” OR “stigma” OR “health related quality of life” OR “bullying” OR “post-traumatic stress” OR “post-traumatic stress disorder.”Concept 3“Pediatric” OR “children” OR “adolescents.”

Each unique abstract was examined by the lead author (T.T.) for inclusion criteria and any that were explicitly unmet (e.g., study of non-humans, adult population) were discarded. The co-authors (L.N., B.D., E.E.) reviewed the selected list for agreement. Reference lists of identified articles and book chapters were also reviewed for additional studies. Unpublished manuscripts, abstracts, case reports, and dissertations were not included. 

## 3. Search Results

Based on the selected search criteria, two studies were identified for review (Table 1). The results of the search find that the only psychosocial domains researched, to date, are anxiety and depression. The two studies suffered from methodological limitations, including retrospective chart review, leaving no prospective data on the rates of anxiety or depression in pediatric GP, nor how these co-morbidities may impact patient outcomes. Therefore, in addition to providing insights into the limited studies to date, we will also provide an overview of these constructs in other pediatric populations and how these may be extrapolated to the pediatric GP patient population.

## 4. Psychological Co-Morbidities

### 4.1. Anxiety and Depression

It is logical that chronic, severe symptoms—whether they be nausea and vomiting, diarrhea, or pain—would cause increased anxiety and depression in the child experiencing them. Due to the possible role of aberrant brain–gut axis signaling involved in GP, increases in anxiety or depression may, in turn, exacerbate symptoms of nausea and pain in these patients [14], leading to a debilitating cycle of digestive symptoms and psychological distress. Only two studies provided some insight into the presence of anxiety and depression in children with GP. The first study, published in 2012, reported rates of anxiety and depression in pediatric patients with GP [15]. In this study, 6.3% experienced anxiety and 4% reported depression, which is considerably lower than adults with GP, where research suggests up to 33% experience significant anxiety and 42% have moderate to severe depression [16]. However, this study was based on a single study utilizing retrospective chart review and should be interpreted with caution. The second study, published in 2014, evaluated children undergoing gastric emptying testing and found that children with GP did not differ significantly in state or trait anxiety when compared to children without GP [17]. However, all patients were undergoing evaluation for chronic nausea, vomiting or other indications for testing, so it is difficult to discern how this comparison is clinically meaningful.

While rates of anxiety and depression in adults with GP are similar to findings among other chronic digestive diseases, the available data suggest rates among children with GP are lower than is found in other digestive disease groups. For example, in children with chronic idiopathic nausea, rates of anxiety are as high as 70% [18]. Comparatively, approximate rates of anxiety in children with functional abdominal pain (FAP) range from 35 to 45%, 16 to 41% in eosinophilic esophagitis (EoE), and 20% in IBD. Similar rates exist for depression: 40–50% for FAP, 9–24% in EoE [19], and 18% in IBD [20]. It is likely that these discrepancies are due to the pediatric GP data being limited to two studies of 339 children with retrospective data collected from the patients’ charts versus a systematic measurement of these symptoms using standardized questionnaires. These limitations underscore the imperative need for additional research into anxiety and depression in these patients. 

### 4.2. Health-Related Quality of Life

Health-related quality of life (HRQOL) is a patient-reported outcome of increasing importance in medical populations. Unfortunately, no studies have evaluated the effects of GP on HRQOL in pediatric patients with the exception of treatment clinical trials with HRQOL as a secondary outcome; limited research does exist evaluating HRQOL in adults with GP [21,22]. However, in these studies, HRQOL was measured with a non-disease-specific scale (i.e., SF-36), so the nuances of the effects of GP on HRQOL are likely not well understood. Despite the lack of GP-specific research, we can extrapolate some risks of poorer HRQOL from similar pediatric populations as to how these might manifest in GP patients. Degradations in HRQOL are consistently found in children with other chronic digestive diseases including functional abdominal pain (FAP) [23] and nausea/vomiting associated with cancer treatment [24]. Lower HRQOL is consistently associated with other poor patient outcomes including more severe symptoms, anxiety, and depression across multiple disease groups. 

### 4.3. Eating Behaviors and Nutritional Assessment

Psychological distress and disordered eating behaviors are often interrelated in patients with chronic digestive conditions. Children with GP most commonly present with vomiting (42–68%) and abdominal pain (35–51%), with 25% experiencing early satiety and weight loss. Based on these data, it is likely they will engage in compensatory behaviors to try to mitigate these symptoms [15]. Little is known about how gastroparesis impacts growth, eating behaviors, and food related quality of life (FRQoL; i.e., the specific effects of dietary modifications and the role of food in digestive symptoms has on day-to-day function) in children with GP. Additionally, food avoidance to compensate for GP symptoms may lead to the worsening of gastric emptying as seen in pediatric patients with anorexia nervosa [25].

An unintended consequence of focusing on food as a means to control digestive symptoms may be the development of avoidant or restrictive eating behaviors, as further restriction and avoidance may be sought to better control symptoms. Robson and colleagues recently described the presentation of avoidant restrictive food intake disorders (ARFID) in children with other chronic GI disorders including EoE [26]. Risk factors for the development of ARFID includes anxiety-based food avoidance, medically directed dietary avoidance or modification, child temperament and the cycle of starvation leading to decreased energy and loss of appetite. This model deserves further study in pediatric patients with GP and especially among those utilizing dietary modification to alter symptoms.

Proper nutritional assessment in children with GP includes both growth and eating behaviors through a diet history and can screen for problematic avoidance. Taking a detailed diet history is an essential element of nutritional assessment and best completed with the help of a registered dietitian. A diet history can allow for the assessment of current nutritional intake as well as eating behaviors including any self-limiting or avoidant behaviors. Patients may adopt restrictions based on inaccurate assumptions about which foods drive symptoms. A nutrition care plan may call for the reintroduction of suspected trigger foods to evaluate tolerance. Alternate sources of nutrition can be developed and offered by the registered dietitian [27].

### 4.4. Post-Traumatic Stress

To date, no studies have evaluated post-traumatic stress disorder (PTSD) symptoms in pediatric or adult patients with GP. Adverse childhood experiences (ACEs) are traumatic events occurring before the age of 18 and are considered an increasing public health crisis. ACEs affect cognitive [28], social and emotional [29,30], financial [31], and physical health [32] domains across the lifespan, with racial and ethnic minorities disproportionally affected [33].

Among these events, medical trauma arises from hospitalizations, surgeries, procedures, or even significant disease symptoms and can manifest in post-traumatic stress (PTS), which has many overlapping symptoms with PTSD [34,35]. Most research in this area exists in pediatric patients with a history of cancers [36,37], accidents including traumatic brain injury [38,39,40,41], or, more generally, intensive care unit hospitalizations [42]. Often, parents of children undergoing intensive medical procedures also report symptoms of PTSD [36,42,43,44], compounding the risk of GP-associated PTS. It is plausible that patients, especially those with severe GP necessitating the use of nasogastric tube feeding or multiple hospitalizations, are at risk for developing PTS. As such, this is a critical area for future research. 

### 4.5. Social Relationships and Stigma

No studies to date have address social relationships and stigma related to pediatric GP. Children with chronic medical conditions are at risk for stigmatization, bullying, and social isolation [45]. These issues are found in children with IBD [46,47] and chronic abdominal pain [48]. Similarly, children with food allergies may experience bullying at twice the rate of their peers without a food allergy [49]. It is likely that children with GP are also at risk of social isolation and bullying due to their condition. While bowel symptoms, such as diarrhea or flatulence, are often associated with being socially taboo, vomiting also has many qualities associated with stigma, including perception of contagion (e.g., a stomach virus) and vomiting’s non-aesthetic qualities [50]. Children who experience vomiting often fear attending school due to social embarrassment. Other traits of GP including it being an invisible illness, potentially having a relapsing and remitting course, and people’s concerns of their own distress when associating with the child with GP lend it to considerable stigmatization [51]. Other DGBIs, including irritable bowel syndrome, are even stigmatized by healthcare professionals, with patients citing considerable concerns that their symptoms are deemed “all in their head” [52]. In adolescent girls, assumptions may be made about the presence of eating disorders such as anorexia or bulimia nervosa, which are also highly stigmatized conditions [53]. All of these factors make research into the stigmatization of children with GP an equally important focus.

## 5. Recommendations for Future Research

The present review identified only two studies in pediatric patients with GP related to psychosocial outcomes. While this is disappointing, it appears consistent with the overall dearth of research on pediatric GP, compounded by the fact existing studies are overwhelmingly retrospective in nature. As such, our recommendations for future research are to conduct prospective cross-sectional and longitudinal studies on the psychosocial domains outlined in this review to align the pediatric GP literature with the more progressed knowledge base for other chronic digestive conditions. Specifically, children with GP report similar symptoms to those with pediatric DGBIs, such as functional dyspepsia. The functional impairment observed in these other patient populations is likely to occur in children with GP, as frequent, intense symptoms are most associated with psychological distress [18,23,54,55,56]. While measuring rates of anxiety and depression is an important step, research into the psychosocial effects of pediatric GP should look beyond these and expand into more nuanced problems such as PTSD/PTS, social isolation, stigmatization and bullying, and transitions from pediatric to adult gastroenterology care. Future studies should seek to investigate the psychological effects of the symptoms of GP (i.e., nausea, vomiting, pain) with comparisons made to pediatric patients with conditions exhibiting similar symptom profiles (i.e., rumination disorder, functional dyspepsia).

In addition to the basic measurement and assessment of correlates of psychosocial issues in pediatric GP, comprehensive evaluation of the efficacy of evidence-based psychogastroenterology interventions, such as cognitive behavioral therapy, gut-directed hypnotherapy, and mindfulness-based therapies in pediatric GP is critical. While many children benefit from pharmacological and surgical approaches, others will remain symptomatic. Psychogastroenterology approaches are effective across the DGBI spectrum and are likely to benefit patients with GP as well. Lastly, the effects of pediatric GP on parents and siblings is also needed to elucidate how this chronic condition affects the family system.

## 6. Conclusions

The current review identified substantial gaps in the research literature on the psychosocial effects of pediatric gastroparesis, with only two studies identified for inclusion. Overall, there is very little research into pediatric GP. While there are more studies among adult patients, the psychosocial research in these patients is also deficient. Compared to other chronic digestive conditions, including DGBIs, IBD, and EoE, research on pediatric GP is considerably underdeveloped, affording ample opportunity for gastroenterologists and health psychologists practicing psychogastroenterology. It is likely that children and adolescents with GP are similar to their peers with other chronic digestive diseases, but also have unique, unidentified and unmet needs.

## Figures and Tables

**Figure 1 children-07-00115-f001:**
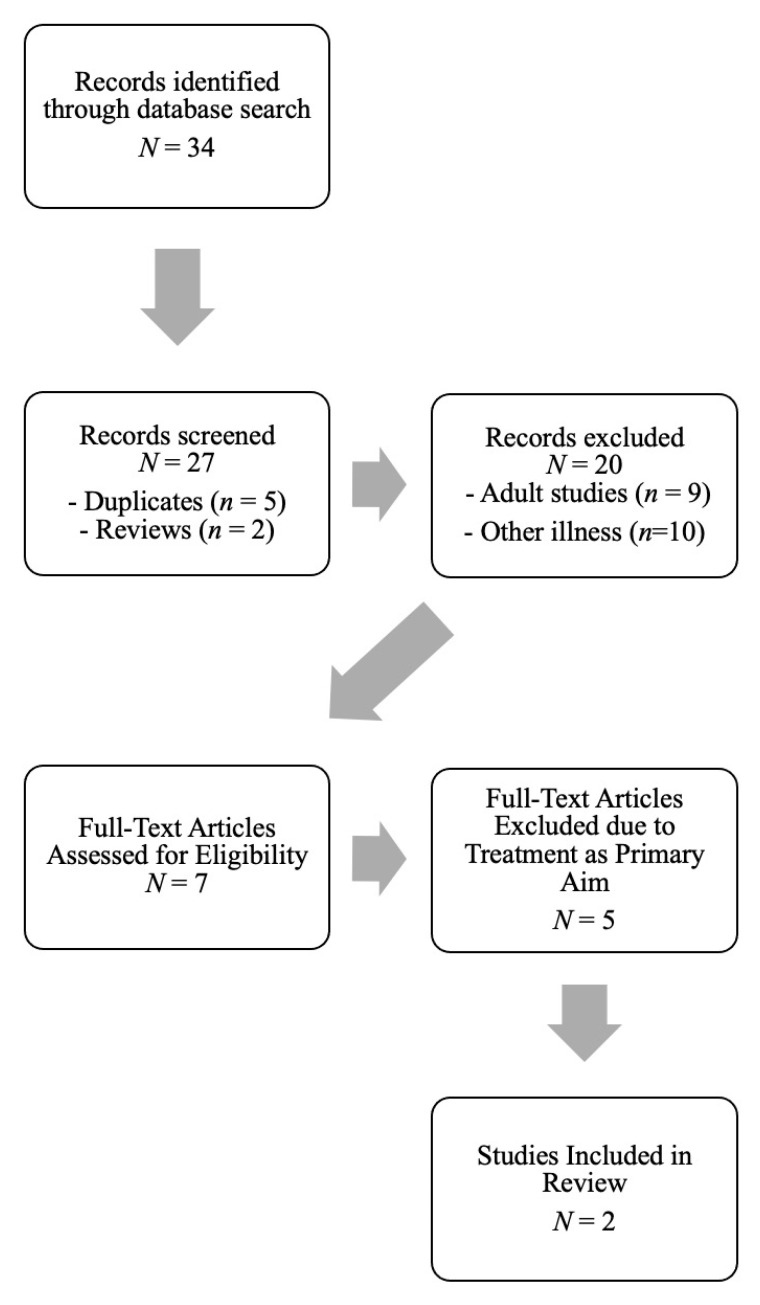
PRISMA Diagram of Search and Article Selection for Final Review.

**Table 1 children-07-00115-t001:** Relevant studies examining psychosocial constructs in pediatric gastroparesis.

	Author Group	Year	Study Design	Sample Size	Constructs Evaluated
1	Wong et al.	2014	Cross-Sectional	100 (25 with GP)	Anxiety
2	Waseem et al.	2012	Retrospective	239	Anxiety, Depression

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
