# Peer review of "The Scarcity of Literature on the Psychological, Social, and Emotional Effects of Gastroparesis in Children"

_children, 2020, doi:10.3390/children7090115_

Round 1

Reviewer 1 Report

Summary:

This review highlights the extreme lack of research into the psychosocial aspects of pediatric gastroparesis. While it is important to highlight this deficiency in the literature, the review only includes 2 articles, one being retrospective. Therefore, this summary does not add much and authors are forced to extrapolate a lot to other conditions.

Authors include gastroparesis, a motility disorder, among functional disorders which is inaccurate.

Title: since there are essentially no findings in this area of pediatric GP, title becomes misleading as if the paper will highlight specific findings. I think title would need to state something like “Scarcity of literature on the psychological, social etc Effects of GP in Children”

Author list states..Nguyen “and” –is there another authorto be included? Assume typo

Abstract

-instead of “neurogastroenterological condition” which is nonspecific, would state “gastrointestinal motility disorder”. The latter term or perhaps “gastric dysmotility disorder” would be better to use throughout paper.

-this can’t really be termed a scoping review given that only 2 studies are included; scoping reviews are generally aimed at summarizing findings from a large body of literature and adheres to almost all methodology that a systematic review does (see: Pham et al. 2014 A scoping review of scoping reviews: advancing the approach and enhancing the consistency) Many scoping reviews also include a formal quality assessment of the literature (see: Daudt HM, van Mossel C, Scott SJ BMC Med Res Methodol. 2013 Mar 23; 13:48.) It seems this article should be termed a review article.

-last sentence: would state “..future studies to address psychosocial outcomes”

Intro

-need to remove discussions of functional GI disorders in relation to gastroparesis which is an upper GI/gastric motility disorder and not part of the Rome criteria for functional disorders. Although there are likely brain-gut signaling involved in GP and certainly can mention this as it related to this article, it’s inaccurate and misleading to start this article with a discussion on functional GI disorders (Drossman, D., Hasler, W.(2016). Rome IV—Functional GI Disorders: Disorders of Gut-Brain Interaction Gastroenterology 150(6), 1257-1261)

-3rdparagraph: mitochondrial disorders probably should not be listed as one of the most common reasons for GP in children. These are still rare and there is limited data on what actually constitutes a mitochondrial disorder---this term is frequently used when a mitochondrial disorder is suspected and GP cases linked to actual mito disorders would not be common. A commonly cited reference (Bhardwaj et al.Impaired gastric emptying and small bowel transit in children with mitochondrial disorders), only evaluated 26 patients with mitochodrial dysfunction. Post-viral and idiopathic are most common causes in children. Drug-induced or post-surgical could be mentioned as other causes rather than mitochondrial disorders. Diabetic GP is rare in children.

-3rdpara: would specify “gastric” electrical stimulation implants

-4thpara; remove part that states “especially when compared to other DGBIs”

Methods

Section 2.1.

-inclusion criteria states only idiopathic or diabetic GP, later in Methods (search keywords), “post-viral” is listed. Hopefully authors included post-viral GP as diabetic is almost non-existent in the pediatric population. Please clarify.

-did authors also exclude abstracts? Would specify.

-HRQOL section: is there any literature in pediatric GP with “treatment trials with HRQL as a secondary outcome”? if so, should cite these. Authors cite adult studies here.

-not sure that extrapolating to IBD and EoE populations are as relevant in this paper, seems a bit out of place. I think authors should consider commenting on any adult GP data on this topic as it would relate much better rather than picking other pediatric diseases.

Section 4.2:

-typo in first sentence (often written twice)

-2ndsentence also need revision; remove “As…” at beginning of sentence

-the comments on proper nutritional assessment is important but don’t think authors need to cite CDC and WHO growth charts. This should be obvious knowledge of any pediatrician and not relevant to include in this paper.

-there is some data on eating disorders in adolescents/young adults with GP; this would be better cited and referred to than other digestive disorders (see for example Heruc  et al. Effects of starvation and short-term refeeding on gastric emptying and postprandial blood glucose regulation in adolescent girls with anorexia nervosa. Am J Physiol Endocrinol Metab. 2018)

Section 4.3:

--why include ‘wheezing’ in a GI paper on gastroparesis? Would remove this.

Section 4.3:

-please change wording “neurological processes involved in GP” to “due to the possible role of aberrant brain-gut axis signaling involved in GP..”  (cite if there is evidence for this claim or is this a postulation by authors?)

-authors base several claims and comparisons on anxiety/depression prevalence in GP on one single retrospective report (Waseem et al). Don’t think there is enough evidence to say that ‘rates of GP is significantly lower than other digestive diseases’. Would instead highlight this data is retrospective and based on a single study and evidence from adults suggest anxiety/depression is highly prevalent in GP + requires further study. Authors do state that these discprepancies are likely due to the limited data—the fact that there is only one study and zero prospective data needs to be highlighted more.

-end of first para: would change “…difficult to discern how this lack of difference…to “..how this comparison..” (since the gastric emptying tests are so poor and questionably able to capture GP, especially in children, this comparison data is not very helpful).

Section 4.4:

-again not sure why authors compare GP to IBD, this seems out of place and if going to compare or extrapolate, would compare to adult GP data instead

-agree with statement that this is a critical area for future research

Conclusion is well written

Author Response

Dear Reviewer,

Thank you for your thoughtful comments on our manuscript. We have addressed each point as follows:

Authors include gastroparesis, a motility disorder, among functional disorders which is inaccurate.

We apologize for this oversight and have corrected any reference in the paper for GP being a FGID.

Title: since there are essentially no findings in this area of pediatric GP, title becomes misleading as if the paper will highlight specific findings. I think title would need to state something like “Scarcity of literature on the psychological, social etc Effects of GP in Children”

We agree! This was a difficult request to fulfill and appreciate the recommendation to change the title.

Author list states..Nguyen “and” –is there another authorto be included? Assume typo

Typo, corrected.

Abstract

-instead of “neurogastroenterological condition” which is nonspecific, would state “gastrointestinal motility disorder”. The latter term or perhaps “gastric dysmotility disorder” would be better to use throughout paper.

Agreed. We have opted to use gastric dysmotility disorder as we feel this most accurately describes GP.

-this can’t really be termed a scoping review given that only 2 studies are included; scoping reviews are generally aimed at summarizing findings from a large body of literature and adheres to almost all methodology that a systematic review does (see: Pham et al. 2014 A scoping review of scoping reviews: advancing the approach and enhancing the consistency) Many scoping reviews also include a formal quality assessment of the literature (see: Daudt HM, van Mossel C, Scott SJ BMC Med Res Methodol. 2013 Mar 23; 13:48.) It seems this article should be termed a review article.

Thank you for this feedback. We have changed this to narrative review.

-last sentence: would state “..future studies to address psychosocial outcomes”

Changed to reflect this comment.

Intro

-need to remove discussions of functional GI disorders in relation to gastroparesis which is an upper GI/gastric motility disorder and not part of the Rome criteria for functional disorders. Although there are likely brain-gut signaling involved in GP and certainly can mention this as it related to this article, it’s inaccurate and misleading to start this article with a discussion on functional GI disorders (Drossman, D., Hasler, W.(2016). Rome IV—Functional GI Disorders: Disorders of Gut-Brain Interaction Gastroenterology 150(6), 1257-1261)

Thank you for this feedback. In our attempts to bridge the basically non-existent literature in pediatric (and adult, frankly) GP we focused too much on DGBIs as a reference point and blurred the lines between these 2 groups of conditions. We have corrected this throughout the paper.

-3rdparagraph: mitochondrial disorders probably should not be listed as one of the most common reasons for GP in children. These are still rare and there is limited data on what actually constitutes a mitochondrial disorder---this term is frequently used when a mitochondrial disorder is suspected and GP cases linked to actual mito disorders would not be common. A commonly cited reference (Bhardwaj et al.Impaired gastric emptying and small bowel transit in children with mitochondrial disorders), only evaluated 26 patients with mitochodrial dysfunction. Post-viral and idiopathic are most common causes in children. Drug-induced or post-surgical could be mentioned as other causes rather than mitochondrial disorders. Diabetic GP is rare in children.

Updated to reflect this comment.

-3rdpara: would specify “gastric” electrical stimulation implants

Updated to reflect this comment.

-4thpara; remove part that states “especially when compared to other DGBIs”

Updated to reflect this comment.

Methods

Section 2.1.

-inclusion criteria states only idiopathic or diabetic GP, later in Methods (search keywords), “post-viral” is listed. Hopefully authors included post-viral GP as diabetic is almost non-existent in the pediatric population. Please clarify.

We did include post-viral. We have updated the criteria to reflect this.

-did authors also exclude abstracts? Would specify.

Abstracts were excluded. We have updated the search criteria to reflect this.

-HRQOL section: is there any literature in pediatric GP with “treatment trials with HRQL as a secondary outcome”? if so, should cite these. Authors cite adult studies here.

We were unable to find any clinical trials that had HRQOL as a secondary outcome. If the reviewer is aware of something, we greatly appreciate any guidance!

-not sure that extrapolating to IBD and EoE populations are as relevant in this paper, seems a bit out of place. I think authors should consider commenting on any adult GP data on this topic as it would relate much better rather than picking other pediatric diseases.

Updated to reflect this comment.

Section 4.2:

-typo in first sentence (often written twice)

Updated to reflect this comment.

-2ndsentence also need revision; remove “As…” at beginning of sentence

Updated to reflect this comment.

-the comments on proper nutritional assessment is important but don’t think authors need to cite CDC and WHO growth charts. This should be obvious knowledge of any pediatrician and not relevant to include in this paper.

Updated to reflect this comment.

-there is some data on eating disorders in adolescents/young adults with GP; this would be better cited and referred to than other digestive disorders (see for example Heruc  et al. Effects of starvation and short-term refeeding on gastric emptying and postprandial blood glucose regulation in adolescent girls with anorexia nervosa. Am J Physiol Endocrinol Metab. 2018)

Updated to reflect this comment.

Section 4.3:

--why include ‘wheezing’ in a GI paper on gastroparesis? Would remove this.

Updated to reflect this comment.

Section 4.3:

-please change wording “neurological processes involved in GP” to “due to the possible role of aberrant brain-gut axis signaling involved in GP..”  (cite if there is evidence for this claim or is this a postulation by authors?)

Updated to reflect this comment.

-authors base several claims and comparisons on anxiety/depression prevalence in GP on one single retrospective report (Waseem et al). Don’t think there is enough evidence to say that ‘rates of GP is significantly lower than other digestive diseases’. Would instead highlight this data is retrospective and based on a single study and evidence from adults suggest anxiety/depression is highly prevalent in GP + requires further study. Authors do state that these discprepancies are likely due to the limited data—the fact that there is only one study and zero prospective data needs to be highlighted more.

Agree 100% We have updated the data reporting to reflect this comment.

-end of first para: would change “…difficult to discern how this lack of difference…to “..how this comparison..” (since the gastric emptying tests are so poor and questionably able to capture GP, especially in children, this comparison data is not very helpful).

Updated to reflect this comment.

Section 4.4:

-again not sure why authors compare GP to IBD, this seems out of place and if going to compare or extrapolate, would compare to adult GP data instead

Updated to reflect this comment. As there is no adult data on PTSD, we did not add any additional information.

-agree with statement that this is a critical area for future research

Conclusion is well written

Reviewer 2 Report

This paper, entitled “The Psychological, Social, and Emotional Effects of Gastroparesis in Children”, aims to review the literature on this topic and to discuss the inherent differences and challenges of this disorder on functioning and mental health. The authors further state that a goal of this paper is to identify potential areas of research with this specific neuro-gastroenterological condition in a pediatric population.

Strengths of this paper include:

  1. The authors provide a comprehensive overview of gastroparesis, including a simplified explanation of the disorder, information related to etiology, testing that is typically conducted to confirm the diagnosis, and known, recommended treatments.
  2. Overall, the introduction section is well-organized, with the ending providing an appropriate segway into the aims of the paper.
  3. The literature review and manner in which studies were included and excluded appears appropriate.

Minor revisions include:

  1. Line 130 is not a complete sentence; perhaps remove the first word, “As”.
  2. Lines 184-185 do not include a complete sentence (sentence starts with “Thus”). Please revise.

Recommendations of things to consider and address, in order of importance:

  1. MAJOR: The authors describe gastroparesis both as a neuro-gastroenterological condition AND a DGBI. As I understand it, the term DGBI is a new term for “functional gastrointestinal disorder”. Gastroparesis is actually not considered a functional GI disorder according to ROME IV. Therefore, line 36 that describes gastroparesis as a DBGI requires revision. It would be appropriate to describe it as a motility disorder, but not a DGBI.
  2. In lines 82 and 82 the authors reference “Figure 1” but there is no figure included in the paper. Please include this figure for review.
  3. Upon review of the literature the authors state that two studies were able to be included based on identified criteria. In this case, and given the low number of studies meeting criteria, it might be beneficial to describe the methods and results of both studies in more detail. Some information on how psychological data was collected for both studies is included in lines 181-185; perhaps this information can be included when describing the studies in greater detail, either in the chart already included or in the text.
  4. The flow of the paper is somewhat hard to follow. It may be more appropriate for the authors to describe how the literature review was conducted, the two studies that were identified, and then describe the studies as recommended above (in point #2). During this explanation it seems appropriate to explain rates of anxiety and depression that were noted in each study rather than wait to mention it in Section 4.3 (starts on line 161). From here it might make more sense for the authors to proceed with explaining that other psychological constructs, particularly those studied in other pediatric gastroenterological conditions, have clearly not been studied in patients diagnosed with gastroparesis, followed by an explanation of the constructs mentioned in the paper under Section 4, Psychological Comorbidities.
  5. It is recommended that the authors consider moving the description of nutrition and dietary interventions for the treatment of gastroparesis (lines 136-160) to after medication treatments are discussed (line 65). The authors can then circle back briefly to this when discussing the possibility of future research assessing avoidant and restrictive eating behaviors in patients diagnosed with gastroparesis.
  6. In the “Recommendations for Future Research” section it may be beneficial to further elaborate on the fact that the symptoms associated with gastroparesis very closely mimic symptoms associated with pediatric DGBIs (most notably functional dyspepsia). Symptom frequency and intensity, in addition to many other factors, contribute to the functional impairment and psychological distress commonly observed in both disorders; therefore, the literature describing the psychological impact of symptoms such as nausea, vomiting and abdominal pain may actually be very relevant. In order to verify this it might be recommended that research focus specifically on the presence and levels of psychological and social distress in subsets of pediatric patients diagnosed with different DGBIs (e.g. functional abdominal pain, rumination disorder, functional dyspepsia, irritable bowel syndrome) and other gastroenterological conditions (e.g. IBD, EoE) in comparison to patients diagnosed specifically with gastroparesis.

Author Response

Dear Reviewer:

Thank you for your thoughtful comments on our manuscript. Please find our response to each recommendation as follows:

This paper, entitled “The Psychological, Social, and Emotional Effects of Gastroparesis in Children”, aims to review the literature on this topic and to discuss the inherent differences and challenges of this disorder on functioning and mental health. The authors further state that a goal of this paper is to identify potential areas of research with this specific neuro-gastroenterological condition in a pediatric population.

Strengths of this paper include:

  1. The authors provide a comprehensive overview of gastroparesis, including a simplified explanation of the disorder, information related to etiology, testing that is typically conducted to confirm the diagnosis, and known, recommended treatments.
  2. Overall, the introduction section is well-organized, with the ending providing an appropriate segway into the aims of the paper.
  3. The literature review and manner in which studies were included and excluded appears appropriate.

Minor revisions include:

  1. Line 130 is not a complete sentence; perhaps remove the first word, “As”.

Updated to correct this typo.

  1. Lines 184-185 do not include a complete sentence (sentence starts with “Thus”). Please revise.

Updated to correct this typo.

Recommendations of things to consider and address, in order of importance:

  1. MAJOR: The authors describe gastroparesis both as a neuro-gastroenterological condition AND a DGBI. As I understand it, the term DGBI is a new term for “functional gastrointestinal disorder”. Gastroparesis is actually not considered a functional GI disorder according to ROME IV. Therefore, line 36 that describes gastroparesis as a DBGI requires revision. It would be appropriate to describe it as a motility disorder, but not a DGBI.

Thank you for noting this oversight on our part. In our attempts to locate literature to discuss, since there is very little in GP, we blurred the lines between DGBI/FGID and GP. We have corrected this throughout the paper.

  1. In lines 82 and 82 the authors reference “Figure 1” but there is no figure included in the paper. Please include this figure for review.

We checked our uploaded files on the original submission and have included the PRISMA diagram.

  1. Upon review of the literature the authors state that two studies were able to be included based on identified criteria. In this case, and given the low number of studies meeting criteria, it might be beneficial to describe the methods and results of both studies in more detail. Some information on how psychological data was collected for both studies is included in lines 181-185; perhaps this information can be included when describing the studies in greater detail, either in the chart already included or in the text.

See point 4.

  1. The flow of the paper is somewhat hard to follow. It may be more appropriate for the authors to describe how the literature review was conducted, the two studies that were identified, and then describe the studies as recommended above (in point #2). During this explanation it seems appropriate to explain rates of anxiety and depression that were noted in each study rather than wait to mention it in Section 4.3 (starts on line 161). From here it might make more sense for the authors to proceed with explaining that other psychological constructs, particularly those studied in other pediatric gastroenterological conditions, have clearly not been studied in patients diagnosed with gastroparesis, followed by an explanation of the constructs mentioned in the paper under Section 4, Psychological Comorbidities.

We have addressed items 3 and 4 together by moving the findings from the 2 studies included under the table rather than discussing them further down.

  1. It is recommended that the authors consider moving the description of nutrition and dietary interventions for the treatment of gastroparesis (lines 136-160) to after medication treatments are discussed (line 65). The authors can then circle back briefly to this when discussing the possibility of future research assessing avoidant and restrictive eating behaviors in patients diagnosed with gastroparesis.

Agreed, this will help flow. We moved the first 3 sentences related to dietary treatment under the treatments section, leaving the remainder of the text in the eating behaviors and assessment section.

  1. In the “Recommendations for Future Research” section it may be beneficial to further elaborate on the fact that the symptoms associated with gastroparesis very closely mimic symptoms associated with pediatric DGBIs (most notably functional dyspepsia). Symptom frequency and intensity, in addition to many other factors, contribute to the functional impairment and psychological distress commonly observed in both disorders; therefore, the literature describing the psychological impact of symptoms such as nausea, vomiting and abdominal pain may actually be very relevant. In order to verify this it might be recommended that research focus specifically on the presence and levels of psychological and social distress in subsets of pediatric patients diagnosed with different DGBIs (e.g. functional abdominal pain, rumination disorder, functional dyspepsia, irritable bowel syndrome) and other gastroenterological conditions (e.g. IBD, EoE) in comparison to patients diagnosed specifically with gastroparesis.

Thank you for these excellent recommendations. We have incorporated these into our future research section.

Round 2

Reviewer 1 Report

Authors have put great effort into revising manuscript and responded appropriately to all the feedback.